# 'We weren't checked in on, nobody spoke to us': an exploratory qualitative analysis of two focus groups on the concerns of ethnic minority NHS staff during COVID-19

Jehanita Jesuthasan [1] Richard A Powell [2] Victoria Burmester [1] Dasha Nicholls [1]

¹Department of Brain Sciences, Imperial College London Faculty of Medicine, London, UK
²Department of Primary Care and Public Health, Imperial College London School of Public Health, London, UK

**Correspondence to**
Dr Dasha Nicholls;
d.nicholls@imperial.ac.uk

## ABSTRACT

**Objective** To gain exploratory insights into the multifaceted, lived experience impact of COVID-19 on a small sample of ethnic minority healthcare staff to cocreate a module of questions for follow-up online surveys on the well-being of healthcare staff during the pandemic.

**Design** A cross-sectional design using two online focus groups among ethnic minority healthcare workers who worked in care or supportive roles in a hospital, community health or primary care setting for at least 12 months.

**Participants** Thirteen healthcare workers (11 female) aged 26–62 years from diverse ethnic minority backgrounds, 11 working in clinical roles.

**Results** Five primary thematic domains emerged: (1) *viral vulnerability*, centring around perceived individual risk and vulnerability perceptions; (2) *risk assessment*, comprising pressures to comply, perception of a tick-box exercise and issues with risk and resource stratification; (3) *interpersonal relations in the workplace*, highlighting deficient consultation of ethnic minority staff, cultural insensitivity, need for support and collegiate judgement; (4) *lived experience of racial inequality*, consisting of job insecurity and the exacerbation of systemic racism and its emotional burden; (5) *community attitudes*, including public prejudice and judgement, and patient appreciation.

**Conclusions** Our novel study has shown ethnic minority National Health Service (NHS) staff have experienced COVID-19 in a complex, multidimensional manner. Future research with a larger sample should further examine the complexity of these experiences and should enumerate the extent to which these varied thematic experiences are shared among ethnic minority NHS workers so that more empathetic and supportive management and related occupational practices can be instituted.

## Strengths and limitations of this study

► This is one of the first studies to examine qualitatively the experiences of ethnic minority healthcare staff during the COVID-19 pandemic.
► Due to the self-selected nature of the sample, the experiences of participants may not be representative of the ethnic minority healthcare workforce as a whole.
► Female healthcare workers were disproportionately represented in our sample.
► Given its exploratory nature, the sample size was potentially insufficient to achieve data saturation and to be representative of the diversity of experiences of ethnic minority staff in the National Health Service.

redeployment to areas outside their professional training[2]—sometimes engaging in tasks that transgress individual moral consciences and values[3]—limited resources and vital medical equipment,[4] and unprecedented patient service demand amid rapidly changing care guidelines.[5] Staff are vulnerable to contracting the virus and at risk of work stress, moral injury and mental ill health, including clinical depression, post-traumatic stress disorder, substance misuse and suicide.[6]

Ethnic minority care staff have been disproportionately affected by the pandemic's clinical impact.[7] The impact is multifaceted: economic, social, attitudinal and cultural, as well as occupational, physical and psychological.[8] [9] Understanding these broader impacts can facilitate a more holistic and empathetic understanding of the lived experiences of COVID-19 among healthcare staff from ethnic minorities, generating greater awareness of health-seeking behaviour in and outside the workplace, and perceived

## INTRODUCTION

COVID-19 has adversely impacted the occupational roles and physical and mental well-being of healthcare staff.[1] Many have experienced disrupted support structures,

impediments to seeking and receiving support.[10] [11] Increased understanding can also inform public health campaigns addressing, for example, disinclination to accept vaccines.[12–14]

The MeCare study in North West (NW) London, collaborating with partners in the larger *NHS Check* study,[15] [16] is longitudinally examining the mental health and well-being of National Health Service (NHS) staff (including general practitioners) in NW London—where 49% of NHS staff are from ethnic minority backgrounds[17]—and exploring how to support staff during pandemics. This paper reports on the cocreation of a module of questions targeted towards staff from ethnic minority backgrounds as part of follow-up online surveys.

## METHOD

### Participants

We explored the experiences of ethnic minority healthcare workers selected through purposive sampling. Participants were recruited through patient and public involvement (PPI) networks and promotional materials circulated among NHS trusts. Participants were adults from an Office for National Statistics[18] ethnic minority background who worked in care or supportive roles in a hospital, community health or primary care setting for at least 12 months.

Before beginning, the purpose of the focus group, the purpose of recording the sessions, the anonymisation of responses for analysis and publication were reiterated. Participants were made aware that the sessions were to be recorded to facilitate subsequent analysis and informed that by attending they were agreeing to the recording of their contribution. The recordings were only to be available to the research team and were to be destroyed after they had been transcribed. Participants were also reminded that they could withdraw their participation at any time from the sessions with no consequences but that their responses up to that withdrawal point would be retained. No participants withdrew from the online meeting or afterwards; however, two participants decided not to participate on the day of the sessions: one because of technology connectivity challenges, the other because of a work emergency. Participants were offered the opportunity to ask any questions before providing their informed consent orally before taking part.

### Data collection

Two 2-hour focus groups of six and seven participants were held online—shown to generate idea diversity comparable to in-person equivalents[19]—using Microsoft Teams (MT) and facilitated by RAP, with participants offered the choice of having their session cameras live or not. VB was present at the beginning of both sessions to introduce the research and JJ was present as a note taker throughout both sessions. Field notes were taken by JJ on key points and the conversations were recorded using MT software, from which manuscripts were produced. Both groups followed the study interview guide (online supplemental file 1), which was reactively adapted, in terms of questioning order and prompts. The interview guide was developed based on existing research and news reports on the experiences of ethnic minority healthcare staff in the UK during the pandemic, and on guidance from an expert team at King's College London and independently reviewed and refined by RAP, VB and JJ. In conducting the group discussions, RAP ensured each participant was able to understand and respond to each question, if they so wished. Participants also had the opportunity to respond to each other's comments and responses.

### Data analysis

Discussions were transcribed verbatim and managed using MT's automatic caption generation software and manually checked for errors. RAP, VB and JJ undertook independent content analysis using the template analysis approach,[20] under a constructivist qualitative research paradigm, combining deductive and inductive analyses. The study team subsequently met to reach consensus on emergent themes and subthemes. To ensure the veracity of data interpretation, the findings and draft survey questions were validated by member checking with a sample of group participants.[21]

### Patient and public involvement

Participants were recruited through PPI representatives and derived separately from the MeCare study's target population, they were not direct study participants. Public partners (see contributors) in the Applied Research Collaboration in NW London provided feedback on data interpretation.

## RESULTS

The final sample included 13 healthcare workers (11 female) with a mean age of 42.7 years (SD=10.3 years). Ethnically, the sample comprised: Eastern European (n=1), Indian (n=2), Mixed White and Black Caribbean (n=1), Caribbean (n=1), Mixed White and Asian (n=1), African (n=2), Arab (n=1), Bangladeshi (n=1), Asian Other (n=2) and Other, in this instance Mixed North African and Eastern European (n=1). Eleven worked in clinical roles in primary and secondary care settings (one GP, three consultants, one nurse, one clinical lead at an urgent care team, two hospital doctors, one pharmacist, one patient housing officer and one psychological well-being practitioner); two worked in administrative roles (one hospital administrator and one assistant improvement manager).

The five primary thematic domains that emerged were: viral vulnerability, risk assessment, interpersonal relationships in the workplace, lived experience of racial inequality and community attitudes (figure 1).

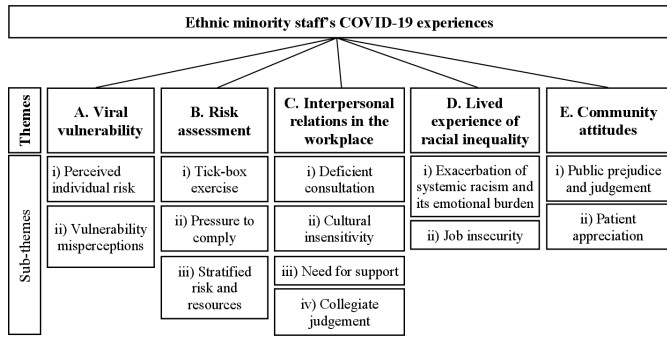

**Figure 1** Themes and subthemes identified through thematic analysis.

## Viral vulnerability

### Perceived individual risk

Participants were sensitive to their increased risk of infection compared with non-ethnic minority colleagues in the same roles. They discussed the impact of this heightened risk on their emotional well-being, reporting personal anxiety and emotional distress.

> I think most upsetting was the initial kind of response and risk stratifications and the deaths of people I knew or knew by indirect means. (P12)

Participants also expressed frustration that individual vulnerability was insufficiently acknowledged within their teams and organisations, a shortcoming one participant suggested was associated with the fact that it was ethnic minorities rather than the non-minority population who were affected.

> If the prevalence and incidence was affecting the White majority population more than the Black population, maybe something could be done differently. (P2)

### Vulnerability misperceptions

Several participants expressed frustration about fundamental misunderstandings regarding the reasons for increased infection risk among ethnic minorities: in public discourse and government reports, differences in viral vulnerability were frequently attributed to individual factors (personal hygiene practices, individual obesity). Participants felt insufficient attention was paid to structural determinants, from exposure differentials to unconscious racism.

> That research was basically saying that [ethnic minorities] are at more risk because they are obese, they have got Type 2 diabetes, they're living in overcrowded situations and all that. […] They did not consider that this could be just because we are exposed more. (P3)

> It's just being aware that it's not entirely biological, the risk that they face. Some of the reasons why we saw people from BAME [Black, Asian and minority

ethnic] background suffer a lot more was because of their social circumstances as well. (P13)

## Risk assessment

### Tick-box exercise

Many participants felt risk assessment exercises had not been treated seriously by their managers and organisations. Although some felt their employer accommodated their identified increased risk, many reported not seeing tangible protective actions for vulnerable ethnic minority staff. By not emphasising the importance of completing risk assessments, and taking minimal action following them, management made staff feel the assessment was a tick-box exercise rather than a considered strategy to reduce staff risk.

> I wasn't able to understand what the point of that risk assessment was and if somebody was found to be high risk population, whether something different was done for that person. (P3)

### Pressure to comply

Despite awareness of ethnic minority communities' COVID-19 vulnerability, participants reported feeling under pressure to continue frontline working. This arose both from managers and an innate sense of duty, with a sense of guilt associated with not seeing in-person patients among higher risk staff members who were shielding and had moved to remote working.

> I did feel quite guilty about not being at work seeing patients. […] Every Thursday for a while [my neighbours] were all standing outside clapping and I didn't go outside. I stayed inside and hid because I felt like a complete fraud. (P11)

Other participants who were still working in person frequently felt they had little choice but to continue working in high-risk positions.

> You kind of feel you're obligated to follow through your duties, but there still is that major risk of putting yourself or your health at risk. (P5)

In terms of extrinsic pressure, the demands on NHS frontline staff limited the alternatives to positioning ethnic minority staff in frontline care provision. Consequently, many reported no choice but to maintain their position on high-risk wards—even when personal protective equipment (PPE) was in short supply—and being pressured to return to work shortly after COVID-19 diagnoses.

> If you pull out all the people on the frontline, then who's going to actually do the work? Who's gonna fill in those gaps? (P5)

> They were desperate to get people on the ground and there was this whole thing about 'it's a respiratory virus and it only lasts 2 weeks max'. So, many

members of staff were harassed, told 'you have to go back, your two weeks are up'. (P12)

Your personal risk doesn't matter, the managers are managers, they're there to get the job done, to apply the pressure coming from above. So, if you're a nurse in a unit that is short-staffed and you turn up to work and they don't have [the necessary PPE], you either get on with your job knowing that you're at high risk, or you become a problem. (P4)

### Stratified risk and resources
Several participants identified flaws in the risk and resultant resources stratification, feeling the organisation's workplace hierarchy was imposed on the risk assessment. They contended the distribution of PPE was based on professional hierarchy rather than individual vulnerability.

The distribution of PPE seems to have followed a hierarchical structure. So, medics are walking around with 3M rubber fit-sealed masks, although they don't spend that much time in COVID areas, while some of the Filipino nurses are struggling to find a 3M mask that fits them. And then when it comes to risk assessing the domestic staff, they're an afterthought. (P4)

### Interpersonal relations in the workplace
### Deficient consultation
The perception of blame for increased vulnerability to COVID-19 being ascribed to ethnic minority communities and a disregard for the role of structural factors fuelled belief in a lack of understanding of ethnic minority experiences. Moreover, the view that attempts to remedy this deficiency were lacking reinforced the sense that larger racial issues were disregarded and thereby unsolved. Participants reported ethnic minorities were excluded from conversations concerning them—such as the construction of risk assessments—and felt the impacts of their vulnerability on their well-being were disregarded within organisations.

We weren't checked in on, nobody spoke to us to see if we were okay, how we felt about it. It just wasn't really made a big deal of. It wasn't something that mattered. (P1)

The issue of not being heard, as somebody from the BAME community. 'Cause I felt like, up to now, nobody's really asked me 'What do you want? What can we do to make you feel protected?' So, I'm aware that the risk assessment was imposed by other people from other organisations. […] It was again that issue of them discussing about us without us. (P2)

### Cultural insensitivity
The disconnect between ethnic and non-ethnic minority staff was associated with difficulties sharing concerns with non-minority supervisors. Several participants cited cultural differences and the associated lack of understanding of ethnic minority staff's experiences, needs and problems as the principal barrier to discussing concerns with non-ethnic minority managers. Indeed, it was suggested that a common background with one's supervisor would facilitate communication and increase feelings of being supported.

I was staying in the trust hotel and went out for a walk […] I was subsequently stopped and searched by police, assaulted. Trying to explain that to my manager resulted in the eye-rolling from them, and them not really understanding the consequences of that. (P4)

It is always much easier to explain to people who are from your own cultural background or somebody who has experienced the same kind of thing rather than actually going to your manager who has got no idea what you're talking about. (P3)

For example, the difficulty obtaining hair caps, needed by ethnic minority staff who do not wash their hair daily, highlighted the impact of under-representation of such staff in senior positions and ethnic minority staff's ability to perform their jobs.

### Need for support
Several participants stated they did not feel adequately supported by their employer and their manager during the pandemic. Some felt particularly unsupported when their supervisor was from a non-ethnic minority background, making it difficult to know their concerns related to ethnicity were heard. For participants who felt they had organisational support, it was seen as improving their well-being and may have mitigated the effects of racial injustices they were increasingly aware of and subjected to during the pandemic.

Will [ethnic minority staff] feel more comfortable and more supported if they have somebody appointed that they can approach if they have any issues? […] Will they feel more comfortable if that person is from the same cultural background? (P3)

I think people were quite protective of me [because I'm ethnic minority] and I personally didn't find that paternalistic or patronising. I felt cared for. (P11)

### Collegiate judgement
Several participants were wary of special treatment being accorded to them, fearing it might negatively impact how they are viewed by non-ethnic minority colleagues. It was suggested some ethnic minority staff may have been reluctant to be risk stratified and made to work from home for fear of being negatively judged by team members.

There were a significant minority of people who, although my recommendation was that they should work from home, they chose to not follow guidance and they wanted to see patients face to face. […] I suspect that might have been because they were worried about their position in the team and about bullying. (P11)

Participants also noted that, in some cases, the special attention given to ethnic minority individuals was experienced as discrimination and a cause for bullying.

> There were a lot of people who were BAME who did feel that there was a spotlight being shone upon them. […] A lot of them didn't feel that that was positive. They felt that it could be seen as quite discriminatory, as them being given special privileges. (P11)

Moreover, several participants were defensive of the reasons for doing their jobs, emphasising their interest in, and passion for, their work and denying ulterior motives, such as hopes to receive special treatment.

> It's work that the BAME group tends to have interest in, and we do it with a love for it, rather than wanting to be specially treated. I think sometimes when there's a lot of talk about BAME groups, it can be a little bit patronising because you feel that's a career that you've always wanted to do. (P9)

This further highlights the fear of one's job performance being judged by colleagues based on ethnicity.

### Lived experience of racial inequality
#### Exacerbation of systemic racism and its emotional burden
There was a strong sense that existing structural workplace inequality and the emotional burden of racial injustice were exacerbated during the pandemic. Participants felt ethnic minority staff have a distinct disadvantage in career advancement, noting they had to work harder than non-minority counterparts to progress their careers. Crucially, several participants felt that increased COVID-19 vulnerability compounded career advancement challenges, and that the pandemic provided an opportunity for unconscious bias among management to manifest in career advancement opportunities given to non-ethnic minority staff.

> I think that there's something similar [to being female] about being BAME compared to being White, where you know you have to work so much harder to get where you want to go to. […] Then there's COVID affecting us in a more serious way, […] it's another disadvantage. (P11)

> There are at least a few cases of people in the same staff group occupying two bandings. And unfortunately, BAME staff at one, slightly lower, band. And management have taken it upon themselves to use the chaos that's going on to promote the same types of people that always get promoted. (P4)

Additionally, participants noted the emotional burden of the pandemic is especially high among ethnic minority staff and that the Black Lives Matter movement compounded feelings of racial injustice evoked in the workplace during the pandemic and highlighted structural, societal problems.

> All of the Black Lives Matter stuff that was going on, and George Floyd and the protests, the whole attitude around them in the workplace was just exacerbating people's feelings and made it very difficult. It felt like you were being attacked from all sides. (P1)

#### Job insecurity
Participants also reported the perception that working from home may threaten their job due to racial inequality in the workplace and that managers were less trusting of ethnic minority staff working from home than non-ethnic minority counterparts.

> I found that the drawback [of working from home] was I wasn't getting my face seen [at work]. […] I actually almost felt that my job was threatened, I became very anxious and very insecure about my place within the team. (P11)

> [At the time I was working from home] I had a colleague I was working with who felt that that pressure [from managers] was given to the ethnic minority group because we were not trusted to be doing what we should be doing. (P9)

### Community attitudes
#### Public prejudice and judgement
The racial injustice experienced by ethnic minority staff in their workplace also occurred in the community, where participants reported witnessing racist slurs against East Asians and anticipating an increase in such attitudes towards themselves.

> I've felt a lot more worried about having sort of racial attacks or racial comments pointed in my direction, given some of the narrative around COVID, such as the 'Chinese Virus'. (P10)

Additionally, the ethnicity-based judgements participants reported receiving from the public reinforced their understanding that ethnic minority staff were seen as less than non-ethnic minority colleagues in society.

> You feel like although you're coming to do something positive but at the same time other people might be making all sorts of judgments as to what you're doing. (P5)

> One of my neighbours said I shouldn't be going outside, so, I said 'No, I work for the NHS' and automatically his response was 'Oh! Are you a cleaner?' (P5)

#### Patient appreciation
Despite racial prejudices experienced in the community, participants noted patients showed an increased recognition of the value ethnic minority communities bring to the NHS and felt this increased patient appreciation was reflective of a broader attitudinal shift towards their key role.

[Patients] were very happy to be seen, happy to be taken care of, they were politer. They really appreciated what we were doing. (P6)

I think there was a shift in [patients'] attitude towards NHS staff in general, which was refreshing. (P1)

In contrast, others were sceptical of whether this recognition was indicative of a real attitudinal shift, recalling past NHS criticism. Moreover, participants highlighted a disconnect between the public's appreciation of the NHS, expressed in weekly street clapping, and practical acknowledgement of the risk to which NHS staff are regularly exposed.

When people would clap for the NHS I would think 'Oh my God they're clapping. What are they talking about? Do they even know what happens inside?' (P13)

## DISCUSSION

This exploratory group process examined the lived experiences of a small sample of healthcare staff from ethnic minority backgrounds during COVID-19 to develop an ethnically empathetic module for follow-up survey stages in the MeCare study (online supplemental file 2). The five primary themes that emerged from our research have highlighted key areas of concern and neglect that need to be enumerated and investigated further to appreciate the extent to which they are shared by ethnic minority staff across the NHS.

These concerns are consistent with research regarding the depression, anxiety and stress associated with perceived COVID-19 risk and the pandemic's impact on ethnic minority communities.[22–24] The perceived pressure to work despite individual risk can significantly impact staff's work performance and mental health. For example, presenteeism—individuals presenting at work but operating suboptimally due to health issues—can have serious consequences if it leads to poor, slow or incorrect decision-making.[25] Moreover, Shah *et al*[26] reported that during the pandemic, ethnic minority staff faced dilemmas around fulfilling their duty and continuing to deliver patient clinical care or exercising mitigating actions to avoid high-risk environments. This is especially relevant given the elevated extrinsic pressure on ethnic minority NHS frontline staff amid surges in COVID-19 cases,[27] combined with staffing shortages[28] and the high proportion of ethnic minority NHS staff.[17]

Ethnic minority staff's mental health is likely to also be impacted by a disconnect between ethnic minority and non-minority staff. Indeed, post-trauma social support from managers influences staff's long-term mental health status,[29 30] and discussion of ethnic minority staff's experiences during COVID-19 with non-minority supervisors can help foster improved attitudes towards mental health in the workplace.[31] The disconnect was manifested in fear of bullying, which is consistent with evidence that ethnic minority staff are more likely than non-minority colleagues to experience staff bullying.[32 33] Additionally, participants' sense that the pandemic exacerbated existing emotional burdens regarding systemic racism is consistent with an interplay between racism and the pandemic.[34] This may also contribute to longer term mental ill health—including the psychological consequences of trauma exposure[29 35 36]—among ethnic minority staff.[37] On a positive note, however, the public's acknowledgement of work undertaken by staff during the pandemic helps promote staff's resilience, thereby protecting their mental health.[38]

Participants' accounts also suggest a perception that nurses and domestic staff—who are more likely to be from ethnic minority backgrounds[39] and are particularly vulnerable to infection due to their physical proximity to patients when providing care[40]—struggled to secure adequate PPE, which was more easily available for senior staff. The perception that PPE distribution was based on professional hierarchy is important. Previous research has found that, among healthcare workers, the incidence of anxiety was highest among non-medical healthcare staff, possibly because they had less first-hand information about the disease and less training on infection control measures and PPE use.[41] Additionally, adequate consultation of ethnic minority staff is especially important given they are already less likely than non-minority counterparts to voice their opinions and raise concerns about inadequate PPE, increasing the likelihood of their needs being overlooked.[33 42]

Racial inequalities were also reported in career advancement and treatment by managers. This is consistent with reports that ethnic minority staff have been restricted to certain NHS roles due to inequalities in career development and workplace and societal discrimination, rendering them more likely to work in critical specialties and services during the pandemic.[43] Moreover, ethnic minority staff are also more likely than non-ethnic minority staff to experience excessive scrutiny and punishment[44] and discrimination[45] from their managers.

In terms of study limitations, an important limitation was that the sample was small and self-selected. As such, the findings may not fully capture the complexity and multifaceted nature of ethnic minority healthcare staff during the COVID-19 pandemic, and participants were likely to feel more strongly about their experience of the pandemic compared with the ethnic minority workforce as a whole. However, participants were purposively sought for their opinions to gain broad insights into this neglected area to inform further inquiry, rather than generating representative data. The MeCare module that this research informed will provide an opportunity to collect data from a broader range of ethnic minority staff and enable exploration of whether these experiences were representative and resonate with ethnic minority staff.

Second, female healthcare workers were disproportionately represented in our sample, potentially omitting

specifically male experiences during the pandemic. Third, there is the possibility that the facilitator's race (non-minority ethnic) biased the data by inhibiting open discussion.[46] However, the facilitator has over 20 years' experience of health research in diverse settings including low and middle-income countries, and JJ, who was present in both focus groups, is minority ethnic. In addition, discussions were frank and unimpeded, and evidence indicates the impacts of the interviewer's ethnicity are minimal.[47] Fourth, the sample size was potentially insufficient to achieve thematic saturation as this was an exploratory exercise to highlight largely unexamined thematic areas that are currently being further researched. To minimise the possibility of the authors' invalid interpretation of the data, the study used respondent validation among a sample of group participants.

Our novel study has clearly shown that ethnic minority NHS staff have experienced COVID-19 in a complex, multidimensional manner. The physical and psychological impacts of the virus have been overlaid by wider occupational, attitudinal (including anticipatory racism) and sociocultural experiences. A need exists to understand the extent to which these varied experiences are shared among ethnic minority NHS workers so more empathetic and supportive management and related occupational practices can be instituted.

**Acknowledgements** We would like to thank the participants for taking part in this research and sharing their experiences.

**Contributors** JJ, RAP, VB and DN conceptualised the study. RAP conducted the focus groups. JJ, RAP and VB analysed the data. JJ wrote the draft with support from RAP, VB and DN. RAP, VB and DN managed the overall design of the study. DN acts as guarantor for the study. Non-author contributors: John Norton and Sandra Jayacodi, partners of the National Institute for Health Research ARC North West London PPI Initiative, verified data interpretation and reviewed the final manuscript.

**Funding** This report is an independent research funded by the National Institute for Health Research North West London Applied Research Collaboration (ARC) and by the Imperial College COVID-19 Research Fund (grant number P88408). NIHR Collaboration for Leadership in Applied Health Research and Care for North West London, London, UK.

**Disclaimer** The views expressed in this publication are those of the authors and not necessarily those of the National Institute for Health Research or the Department of Health and Social Care.

**Competing interests** None declared.

**Patient consent for publication** Not applicable.

**Ethics approval** The MeCare study received ethical approval from the Health Research Authority (IRAS Project ID 290383), which included permission to cocreate a module addressing questions specifically for ethnic minority populations with input from patient and public involvement (PPI) colleagues, project partners and other experts by experience. Participants gave their informed consent before taking part.

**Provenance and peer review** Not commissioned; externally peer reviewed.

**Data availability statement** Data are available upon reasonable request. No data are available.

**ORCID iDs**
Jehanita Jesuthasan http://orcid.org/0000-0002-0341-4546
Richard A Powell http://orcid.org/0000-0003-4968-3714
Victoria Burmester http://orcid.org/0000-0001-7566-3640
Dasha Nicholls http://orcid.org/0000-0001-7257-6605

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
