## [Reviewer comments · BMJ Open]

ARTICLE DETAILS

TITLE (PROVISIONAL)	"We weren't checked in on, nobody spoke to us": An exploratory qualitative analysis of two focus groups on the concerns of ethnic minority NHS staff during COVID-19
AUTHORS	Jesuthasan, Jehanita; Powell, Richard A; Burmester, Victoria; Nicholls, Dasha

VERSION 1 – REVIEW

REVIEWER	Braquehais, María Dolores Galatea Clinic. Galatea Foundation, Inpatient Unit. Integral Care Program for Sick Health Professionals
REVIEW RETURNED	07-Jun-2021

GENERAL COMMENTS	I think this paper addresses a very important neglected issue: the impact of the COVID pandemic in BAME healthcare workers. Its qualitative design enriches the comprehension of the phenomenon. However, there are serious concerns that the authors may consider: 1) The study points to a very ambitious aim and it should be more focused. In that sense, I suggest that the title should be more brief and less "sensationalist": "Understanding ethnic minority NHS staff concerns during COVID-19: a qualitative analysis of some healthcare workers experiences".2) The main (and important) limitation of the study is that the sample is self-selected and only 13 healthcare workers were recruited. Despite of the fact that the information analysis was correct, it seems the conclusions are too ambitious. Therefore, it could be presented as an initial study that may draw the attention of both institutions and healthcare workers on this topic and suggest that this study should be replicated in other settings and with more HCWs to see if their conclusions can be generalized to the NHS:3) Due to the limitations of the study (related to the limited generalizability of the results), the authors should reconsider the future implications of the study. It seems a single study with this limitations should be only the first step when trying to adapt a survey to include some of the themes identified during the analysis. Therefore, the authors should be more cautious with respect to the study implications and consider it both in the title, abstract and along the manuscript.
---

REVIEWER	Torales, Julio National University of Asuncion, Psychiatry
REVIEW RETURNED	07-Jun-2021

GENERAL COMMENTS	This article is scientifically relevant in the current context. The chosen design is also appropriate for the purposes of the research. I recommend that, within the limitations of the study, mention be made of the overrepresentation of women in the sample and whether this could affect the results obtained.
---

REVIEWER	Pandya, Apurva-kumar Indian Institute of Public Health Gandhinagar, Regional Resource Centre for Health Technology Assessment
REVIEW RETURNED	16-Aug-2021

GENERAL COMMENTS	I congratulate the authors for conducting a timely and important study. I have a few observations and comments for strengthening the presentation of the study outcomes.  1. P. 7: The sentence “Participants were a sample of the MeCare study’s target population and recruited through PPI representatives. JN and SJ are public partners of the Applied Research Collaboration in NW London and provided feedback on data interpretation” needs to be clarified; are there study participants involved in the design and implementation of the study? Do authors plan to inform the research participants of this research's results? And if yes, how? 2. It is unclear which was qualitative research paradigm adapted for the study. What methodological orientation was stated to underpin the study? Although ticked in COREQ, adequate details are not provided in the manuscript. 3. There is no description of the focus group interview guide preparation, how the tool was finalized (was it pre-tested), how group interviews were conducted. There was a mention use of an online platform but no information about how many participants, who else remained present from the study team (who was a moderator and a note-taker), were conversations recorded? 4. The issue of informed consent was not mentioned. Details of how many denied participation in the study were not provided. It is important to know whether participants were allowed to deny or withdraw from the study. 5. Description of job role would have provided their vulnerability to contracting COVID and perceived risk. Were all in a similar level of job – officer level, supervisor or managerial, attendants, nurse assistant, or clinical assistants? 6. Also, a description of the gender of study participants and studying gender differences, if any, in experiences would be valuable. 7. In the sample, there were no non-ethnic minority participants. How was the following statement at P. 8 derived? “Participants were sensitive to their increased risk of infection compared to non-ethnic minority colleagues in the same roles”? 8. The sentence at P. 10 “...guilt associated with not seeing in-person patients” is quite incongruent as staff are assumed to see in-person patients. Here details about job description can provide better understanding as many would be working directly with patients and many behind the curtain and their risk perception and psychological experience would vary. 9. The sentence at p.11, “Participants frequently felt they had a minimal choice but to continue working in high-risk positions,” contradicts the earlier statement. Kindly provide the background of job description of all participants.
---

	10. A pseudonym and job role should accompany each verbatim to make sense of participants' reflections. 11. In many instances, authors use in-text citations in the results section (for example, p. 10, line no. 29; p.11, line no. 31-36), which was unnecessary for the results sections. These can be discussed thoroughly in the discussion section. 12. P. 40, item no. 27 What software, if applicable, was used to manage the data? The authors have mentioned the use of Software but mentioned N/A. Either statement in the manuscript needs to be explicit or need to change response in COREQ.
--	---

VERSION 1 – AUTHOR RESPONSE

Reviewer 1:

1) The study points to a very ambitious aim and it should be more focused. In that sense, I suggest that the title should be more brief and less “sensationalist”: “Understanding ethnic minority NHS staff concerns during COVID-19: a qualitative analysis of some healthcare workers experiences”.

As stated above, we have revised the manuscript title to meet this guidance.

The quote embedded into the text is only one illustrative of the general sentiment of study participants. However, if, in the interests of space, it needs to be omitted, then we agree to that action.

2) The main (and important) limitation of the study is that the sample is self-selected and only 13 healthcare workers were recruited. Despite of the fact that the information analysis was correct, it seems the conclusions are too ambitious. Therefore, it could be presented as an initial study that may draw the attention of both institutions and healthcare workers on this topic and suggest that this study should be replicated in other settings and with more HCWs to see if their conclusions can be generalized to the NHS.

We acknowledge the point raised by the reviewer. Ultimately, of course, on the one hand, all study participants are self-selecting, in the sense that they agree for whatever reason to participate in a study. In our study, participants were purposively selected as individuals who would be able to provide insights into potential thematic issues that arose during the course of the pandemic. However, we emphasise this limitation in the conclusions by stating explicitly: ‘An important limitation was that the sample was self-selected and, as such, participants were likely to feel more strongly about their experience of the pandemic compared to the ethnic minority workforce as a whole.’ Adequate Sample size in qualitative research is a topic of academic debate (Vasileiou, K. et al. Characterising and justifying sample size sufficiency in interview-based studies: systematic analysis of qualitative health research over a 15-year period. *BMC Med Res Methodol* 18, 148 (2018)). The numbers in our study, while relatively small, are comparable to many qualitative studies and sufficient to provide initial insights into some of the more important issues experienced during COVID-19. As we state in our conclusions, ‘participants were purposively sought for their opinions to gain broad insights into this neglected area to inform further inquiry, rather than generating representative data.’ Moreover, the fact that our initial insights had validity were shown when we informally compared our initial findings with those from a larger qualitative investigation of the area by another London-based university, with clear overlap revealed between the two studies.

We used the term ‘exploratory’ on four occasions to describe the study to acknowledge the “initial study” element to the work. However, we agree that this “initial study” dimension could be emphasised further and have done so in the section on study limitations with the following sentence: ‘Further research could explore if these issues resonate among ethnic minority staff working and living in different settings, quantify the extent of that resonance through survey research.’

3) Due to the limitations of the study (related to the limited generalizability of the results), the authors should reconsider the future implications of the study. It seems a single study with this limitations should be only the first step when trying to adapt a survey to include some of the themes identified during the analysis. Therefore, the authors should be more cautious with respect to the study implications and considerations both in the title, abstract and along the manuscript.

As stated above, we have now amended the paper so that 1) the future implications of the study (in part based on the limited generalizability of the results) are clearer and 2) edited the title so it is clearer and also reflects the format preferred by BMJ Open.

The reviewer will be aware of the extensive use of the focus groups methodology in developing survey tools (e.g., O'Brien K. Using focus groups to develop health surveys: An example from research on social relationships and AIDS-preventive behaviour. *Health Education Quarterly* 1993;20:361-372; Powell RA, Single HM. Focus groups. *Int J Qual Health Care* 1996;8:499-504) and of enhancing their validity (Powell RA, Single HM, Lloyd KR. Focus groups in mental health research: Enhancing the validity of user and provider questionnaires. *International Journal of Social Psychiatry* 1996;42:193-206), a body of established work to which this paper—with its acknowledged limitations—seeks to add to. On that basis, we hope these changes are satisfactory and meet the requests of the reviewer.

Reviewer: 2

1. This article is scientifically relevant in the current context. The chosen design is also appropriate for the purposes of the research. I recommend that, within the limitations of the study, mention be made of the overrepresentation of women in the sample and whether this could affect the results obtained.

We have, as outlined above, accordingly edited the limitations section of the paper by adding the following text:

'Female healthcare workers were disproportionately represented in our sample, potentially omitting specifically male experiences during the pandemic.'

Reviewer: 3

1. P. 7: The sentence "Participants were a sample of the MeCare study's target population and recruited through PPI representatives. JN and SJ are public partners of the Applied Research Collaboration in NW London and provided feedback on data interpretation" needs to be clarified; are there study participants involved in the design and implementation of the study? Do authors plan to inform the research participants of this research's results? And if yes, how?

No study participants were involved in the design and implementation of the study. However, the sentence has been clarified to:

"Participants were recruited through PPI representatives and derived separately from the MeCare study's target population, they were not direct study participants."

In terms of the validation of the results, yes, research participants were informed of the research results, as reported:

"To minimize the possibility of the authors' invalid interpretation of the data, the study used respondent validation among a sample of group participants."

As for the wider dissemination of the results: we have reported back on the baseline phase of the study (which does not include this work with people from ethnic minority backgrounds) via our contact people (CEOs, communications leads and heads of HR) among the participating study sites, discussing options for optimal dissemination to all participants (e.g., via staff workforce emails, Twitter links and staff newsletters). When we have the three-month follow-up data (which incorporates the

questions developed from this paper), we will develop a research uptake strategy that explicitly includes platforms and outlets for contacting especially staff from ethnic minority backgrounds.

2. It is unclear which was qualitative research paradigm adapted for the study. What methodological orientation was stated to underpin the study? Although ticked in COREQ, adequate details are not provided in the manuscript.

We thank the reviewer for this thoughtful question and have therefore edited the data analysis section text to read:

“RAP, VB and JJ undertook independent content analysis using the template analysis approach, under a constructivist qualitative research paradigm, combining deductive and inductive analyses.”

3. There is no description of the focus group interview guide preparation, how the tool was finalized (was it pre-tested), how group interviews were conducted. There was a mention use of an online platform but no information about how many participants, who else remained present from the study team (who was a moderator and a note-taker), were conversations recorded?

We have amended the text accordingly to ensure greater clarity for readers:

“The interview guide was developed based on existing research and news reports on the experiences of ethnic minority healthcare staff in the UK during the pandemic. While not formally tested prior to conducting the focus groups, the guide was based upon guidance from an expert team at King’s College London and independently reviewed and refined by RP, VB and JJ. In conducting the group discussions, RAP ensured each participant was able to understand and respond to each question, if they so wanted. Participants also had the opportunity to respond to each other’s comments and responses.”

To clarify how the group discussions were conducted, we sought to clarify the text again:

“Two 2-hour focus groups of 7 and 6 participants were held online—shown to generate idea diversity comparable to in-person equivalents¹⁹—using Microsoft Teams (MT) and facilitated by RAP, with participants offered the choice of having their session cameras live or not. VB was present at the beginning of both sessions to introduce the research and JJ was present as a note-taker throughout both sessions. Field notes were taken by JJ on key points and the conversations were recorded using MT’s software, from which manuscripts were produced.”

4. The issue of informed consent was not mentioned. Details of how many denied participation in the study were not provided. It is important to know whether participants were allowed to deny or withdraw from the study.

We did not elaborate upon the issue of informed consent in the main body of the text as we were very conscious of the word limit for the paper. However, we have now expanded upon this issue to read:

“Before beginning, the purpose of the focus group, the purpose of recording the sessions, the anonymisation of responses for analysis and publication were reiterated. Participants were made aware that the sessions were to be recorded to facilitate subsequent analysis and informed that by attending they were agreeing to the recording of their contribution. The recordings were only to be available to the research team and were to be destroyed after they had been transcribed. Participants were also reminded that they could withdraw their participation at any time from the sessions with no consequences but that their responses up to that withdrawal point would be retained. No participants withdrew from the online meeting or afterwards; however, two participants decided not to participate on the day of the sessions: one because of IT connectivity challenges, the other because of a work emergency. Participants were offered the opportunity to ask any questions before providing their informed consent orally before taking part”.

5. Description of job role would have provided their vulnerability to contracting COVID and perceived risk. Were all in a similar level of job – officer level, supervisor or managerial, attendants, nurse assistant, or clinical assistants?

The sample was comprised of participants from a range of job roles and seniorities (i.e., broadly representative), including a GP, clinical lead at an urgent care team, consultants and a pharmacist). We were concerned, however, that linking individuals to given responses could compromise their promised anonymity. As an indication of the variability in the study sample, however, we have edited the text to now read:

“Eleven worked in clinical roles in primary and secondary care settings (one GP, three consultants, one nurse, one clinical lead at an urgent care team, two hospital doctors, one pharmacist, one patient housing officer, and one psychological wellbeing practitioner); two worked in administrative roles (one hospital administrator and one assistant improvement manager).”

6. Also, a description of the gender of study participants and studying gender differences, if any, in experiences would be valuable.

As mentioned on p.8, the sample included 11 females and 2 males. The low numbers do not permit reasonable inter-gender comparative analyses. However, the limitation of having a disproportionate number of females in the sample is now noted:

‘Female healthcare workers were disproportionately represented in our sample, potentially omitting specifically male experiences during the pandemic.’

7. In the sample, there were no non-ethnic minority participants. How was the following statement at P. 8 derived? “Participants were sensitive to their increased risk of infection compared to non-ethnic minority colleagues in the same roles”?

This sentence does not refer to differences in sensitivity to or awareness of the risk between individuals of different ethnic groups but rather to the statements made by the participants when comparing the risk of infection and severe illness from COVID-19 between ethnic minorities and non-minorities.

8. The sentence at P. 10 “...guilt associated with not seeing in-person patients” is quite incongruent as staff are assumed to see in-person patients. Here details about job description can provide better understanding as many would be working directly with patients and many behind the curtain and their risk perception and psychological experience would vary.

We appreciate the point made around possible incongruities and have therefore sought to clarify the point made:

“with a sense of guilt associated with not seeing in-person patients among higher-risk staff members who were shielding and had moved to remote working.”

9. The sentence at p.11, “Participants frequently felt they had a minimal choice but to continue working in high-risk positions,” contradicts the earlier statement. Kindly provide the background of job description of all participants.

We appreciate the point made around possible contradictions and have therefore sought to clarify the point made:

“Other participants who were still working in person frequently felt they had little choice but to continue working in high-risk positions.”

We have edited the text to include the job description of all participants:

“Eleven worked in clinical roles in primary and secondary care settings (one GP, three consultants, one nurse, one clinical lead at an urgent care team, two hospital doctors, one pharmacist, one patient housing officer, and one psychological wellbeing practitioner); two worked in administrative roles (one hospital administrator and one assistant improvement manager).”

10. A pseudonym and job role should accompany each verbatim to make sense of participants' reflections.

As stated above, the sample was comprised of participants from a range of job roles and seniorities (i.e., broadly representative), including a GP, clinical lead at an urgent care team, consultants, and a pharmacist). We were concerned, however, that linking individuals to given responses could compromise their anonymity.

11. In many instances, authors use in-text citations in the results section (for example, p. 10, line no. 29; p.11, line no. 31-36), which was unnecessary for the results sections. These can be discussed thoroughly in the discussion section.

The references in the results section were used to cite articles in support of the “current events” that the participants referred to (e.g., surges in COVID cases, staffing shortages, and Black Lives Matter). The relevant references have now been moved to the discussion section with the numbering sequence adjusted accordingly.

12. P. 40, item no. 27 What software, if applicable, was used to manage the data? The authors have mentioned the use of Software but mentioned N/A. Either statement in the manuscript needs to be explicit or need to change response in COREQ.

We were initially uncertain if using MTs’ automatic caption generation software could be classified as data management. However, we have since removed the inconsistency between the manuscript at that recorded in the COREQ questionnaire so both reflect that we used, with the text in the paper now reading:

“Discussions were transcribed verbatim and managed using MTs’ automatic caption generation software and manually checked for errors.”

VERSION 2 – REVIEW

REVIEWER	Braquehais, María Dolores Galatea Clinic. Galatea Foundation, Inpatient Unit. Integral Care Program for Sick Health Professionals
REVIEW RETURNED	30-Sep-2021

GENERAL COMMENTS	It is a very interesting study. However, the main objection is that 2 focus group and such a small sample size are not enough to draw conclusions about the complexity and multifaceted nature of ethnic minority NHS staff concerns during the COVID-19. This should remain clear at the title, abstract and discussion. The main limitation, therefore, it is not self-referral (this selection bias would affect the composition of the focus groups) but the very ambitious aim of the study. If the authors include this perspective, it is a valuable. To summarize this reflection, the title could be: "An exploratory qualitative analysis of experiences reported by ethnic minority NHS staff in two online focus groups during COVID-19".
---

	After this new perspective, I recommend the authors to rewrite the abstract, the aim and the discussion sections.
REVIEWER	Pandya, Apurva-kumar Indian Institute of Public Health Gandhinagar, Regional Resource Centre for Health Technology Assessment
REVIEW RETURNED	03-Oct-2021
GENERAL COMMENTS	Authors have incorporated reviewers' comments. Manuscript has come out very well and can be published.

VERSION 2 – AUTHOR RESPONSE

We thank the reviewer for their feedback and agree that the study's small sample size relative to its aim is an important limitation. We have made the following changes to include this perspective:

We have changed the title to better reflect the small sample size, although we have adapted the reviewer's suggested title to be more concise and still include the quote which we feel is valuable:

" "We weren't checked in on, nobody spoke to us": An exploratory qualitative analysis of two focus groups on the concerns of ethnic minority NHS staff during COVID-19"

We have also edited the abstract to emphasise the exploratory nature of the study and its aim of examining the experiences of a small sample of ethnic minority NHS staff. The objective is now:

"To gain exploratory insights into the multi-faceted, lived-experience impact of COVID-19 on a small sample of ethnic minority healthcare staff to co-create a module of questions for follow-up online surveys on the wellbeing of healthcare staff during the pandemic."

We highlight the need for further work with a larger sample in order to draw stronger conclusions about the concerns of ethnic minority healthcare staff, given the multifaceted nature of their experiences, in the conclusion of the abstract, stating:

"Future research with a larger sample should further examine the complexity of these experiences"

We also addressed the limited representativeness of our study due to its sample size in the Article Summary, adding that the sample was potentially too small "to be representative of the diversity of experiences of ethnic minority staff in the NHS."

Finally, we edited the limitations section of the discussion, adding that, as a result of the small sample, "the findings may not fully capture the complexity and multi-faceted nature of ethnic minority healthcare staff during the COVID-19 pandemic".